# LEARNING A SPATIO-TEMPORAL EMBEDDING FOR VIDEO INSTANCE SEGMENTATION

## ABSTRACT

Understanding object motion is one of the core problems in computer vision. It requires segmenting and tracking objects over time. Significant progress has been made in instance segmentation, but such models cannot track objects, and more crucially, they are unable to reason in both 3D space and time. We propose a new spatio-temporal embedding loss on videos that generates temporally consistent video instance segmentation. Our model includes a temporal network that learns to model temporal context and motion, which is essential to produce smooth embeddings over time. Further, our model also estimates monocular depth, with a self-supervised loss, as the relative distance to an object effectively constrains where it can be next, ensuring a time-consistent embedding. Finally, we show that our model can accurately track and segment instances, even with occlusions and missed detections, advancing the state-of-the-art on the KITTI Multi-Object and Tracking Dataset.

## 1 INTRODUCTION

Explicitly predicting the motion of actors in a dynamic scene is a critical component of intelligent systems. Humans can seamlessly track moving objects in their environment by using cues such as appearance, relative distance, and temporal consistency. The world is rarely experienced in a static way: motion (or its absence) provides essential information to understand a scene. Similarly, incorporating past context through a temporal model is essential to segment and track objects consistently over time and through occlusions.

From a computer vision perspective, understanding object motion involves segmenting instances, estimating depth, and tracking instances over time. Instance segmentation, which requires segmenting individual objects at the pixel level, has gained traction with challenging datasets such as COCO (Lin et al., 2014), Cityscapes (Cordts et al., 2016) and Mapillary Vistas (Neuhold et al., 2017). Such datasets, which only contain single-frame annotations, do not allow the training of video models with temporally consistent instance segmentation, nor does it allow self-supervised monocular depth estimation, that necessitates consecutive frames. Yet, navigating in the real-world involves a three-dimensional understanding of the other agents with consistent instance segmentation and depth over time. More recently, a new dataset containing video instance segmentation annotations was released, the KITTI Multi-Object and Tracking Dataset (Voigtlaender et al., 2019). This dataset contains pixel-level instance segmentation on more than 8,000 video frames which effectively enables the training of video instance segmentation models.

In this work, we propose a new spatio-temporal embedding loss that learns to map video-pixels to a high-dimensional space[1]. This space encourages video-pixels of the same instance to be close together and distinct from other instances. We show that this spatio-temporal embedding loss, jointly with a deep temporal convolutional neural network and self-supervised depth loss, produces consistent instance segmentations over time. The embedding accumulates temporal context thanks to the temporal model, as otherwise, the loss would only be based on appearance. The temporal model is a causal 3D convolutional network, which is only conditioned on past frames to predict the current embedding and is capable of real-time operation. Finally, we show that predicting depth improves the quality of the embedding as knowing the distance to an instance constrains its future location given that objects move smoothly in space.

---

[1]See a video demo of our model here: https://youtu.be/pqRPXRUlQ2I

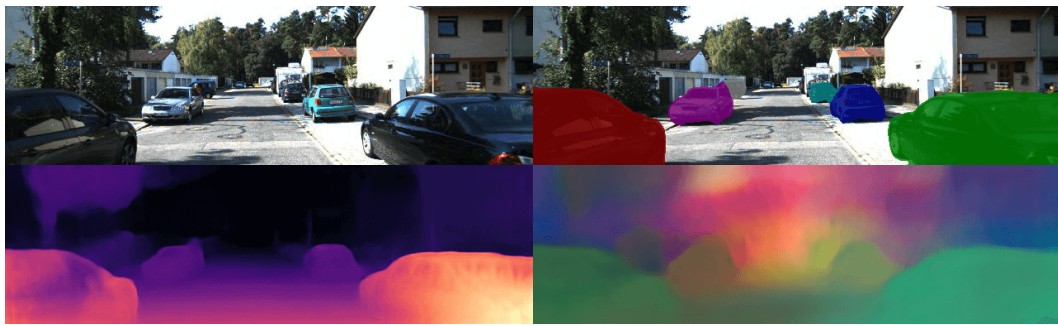

Figure 1: An illustration of our video instance segmentation model. Clockwise from top left: input image, predicted multi-object instance segmentation, visualisation of the high-dimensional embedding and predicted monocular depth.

To summarise our novel contributions, we:

- introduce a new spatio-temporal embedding loss for video instance segmentation,
- show that having a temporal model improves embedding consistency over time,
- improve how the embedding disambiguates objects with a self-supervised monocular depth loss,
- handle occlusions, contrary to previous IoU based instance correspondence.

We demonstrate the efficacy of our method by advancing the state-of-the-art on the KITTI Multi-Object and Tracking Dataset (Voigtlaender et al., 2019). An example of our model's output is given by Section 1.

## 2 RELATED WORK

Two main approaches exist for single-image instance segmentation: region-proposal based (He et al., 2017; Hu et al., 2018; Chen et al., 2018; Liu et al., 2018) and embedding based (Brabandere et al., 2017; Kong & Fowlkes, 2018; Fathi et al., 2017; Kendall et al., 2018). The former method relies on a region of interest proposal network that first predicts bounding boxes then estimates the mask of the object inside that bounding box. With such strategy, one pixel can belong to the overlap of many bounding boxes, and it is largely unclear how correspondence between pixels can be learned. We instead favour the embedding based method and extend it to space and time.

Capturing the inter-relations of objects using multi-modal cues (appearance, motion, interaction) is difficult, as showcased by the Multi-Object Tracking (MOT) challenge (Xiang et al., 2015). MOT's goal is to infer the trajectories of objects and cover a wide range of applications such as biology (birds (Luo et al., 2014), fish (Spampinato et al., 2008), robot navigation (Elfes, 1989)) and autonomous driving (Petrovskaya & Thrun, 2009; Ess et al., 2009)). Sadeghian et al. (2017) and Son et al. (2017) learned a representation of objects that follows the "tracking-by-detection" paradigm where the goal is to connect detections across video frames by finding the optimal assignment of a graph-based tracking formulation (i.e. each detection is a node, and an edge is the similarity score between two detections).

Collecting large-scale tracking datasets is necessary to train deep networks, but that process is expensive and time-consuming. Vondrick et al. (2018) introduced video colourisation as a self-supervised method to learn visual tracking. They constrained the colourisation problem of a grayscale image by learning to copy colors from a reference frame, with the pointing mechanism of the model acting as a tracker once it is fully trained. The colourisation model is more robust than optical flow based models, especially in complex natural scenes with fast motion, occlusion and dynamic backgrounds.

Voigtlaender et al. (2019) extended the task of multi-object tracking to multi-object tracking and segmentation (MOTS), by considering instance segmentations as opposed to 2D bounding boxes. Motivated by the saturation of the bounding box level tracking evaluations (Pont-Tuset et al., 2017), they introduced the KITTI MOTS dataset, which contains pixel-level instance segmentation on more than 8,000 video frames. They also trained a model which extends Mask R-CNN (He et al., 2017) by

incorporating 3D convolutions to integrate temporal information, and the addition of an association head that produces an association vector for each detection, inspired from person re-identification (Beyer et al., 2017). The temporal component of their model, however, is fairly shallow (one or two layers), and is not causal, as future frames are used to segment past frames. More recently, Yang et al. (2019) collected a large-scale dataset from short YouTube videos (3-6 seconds) with video instance segmentation labels, and Hu et al. (2019) introduced a densely annotated synthetic dataset with complex occlusions to learn how to estimate the spatial extent of objects beyond what is visible.

## 3 Embedding-Based Video Instance Segmentation Loss

Contrary to methods relying on region proposals (He et al., 2017; Chen et al., 2018), embedding-based instance segmentation methods map all pixels of a given instance to a high dimensional space with desirable properties. This overcomes several limitations of region-proposal methods. Firstly, two objects may share the same bounding box and in that situation, it is ambiguous which object mask the model should segment. Secondly, pixels can belong to two separate objects as each prediction is done independently. Finally, the number of detected objects is limited by the fixed number of proposals of the network.

We propose a spatio-temporal embedding loss that extends Brabandere et al. (2017)'s instance embedding loss to video: each pixel belonging to a given instance in space and time is transformed into a unique location in a high dimensional space, using cues such as appearance, context and motion. More concretely, three terms are used in the loss: the attraction loss (Equation (1)) to ensure pixels from the same instance are close to each other, the repulsion loss (Equation (2)) to ensure two separate instances are far from each other and a regularisation term (Equation (3)) so that instance centers should not diverge too much from the origin.

Let us denote the number of instances, $K$, and the subset of indices, $S_k$, corresponding to all the pixels belonging to instance $k$ in the video. $\forall i \in S_k$, $y_i$ is the embedding for pixel position $i$ and $\mu_k$ is the mean embedding of instance $k$: $\mu_k = \frac{1}{|S_k|} \sum_{i \in S_k} y_i$.

$$\mathcal{L}_a = \frac{1}{K} \sum_{k=1}^{K} \frac{1}{|S_k|} \sum_{i \in S_k} \max(0, \|\mu_k - y_i\|_2 - \rho_a)^2 \tag{1}$$

$$\mathcal{L}_r = \frac{1}{K(K-1)} \sum_{k_1 \neq k_2} \max(0, 2\rho_r - \|\mu_{k_1} - \mu_{k_2}\|_2)^2 \tag{2}$$

$$\mathcal{L}_{reg} = \frac{1}{K} \sum_{k=1}^{K} \|\mu_k\|_2 \tag{3}$$

Where $\rho_a$ denotes the attraction radius within a cluster: we want the embedding to be within $\rho_a$ of the centroid. $2\rho_r$ denotes the repulsion radius: we want the centroids of two different clusters to be at least $2\rho_r$ apart. Therefore, if we set $\rho_r > 2\rho_a$, a given pixel embedding of a cluster will be closer to all the pixel embeddings of its cluster than any other pixel embedding.

The spatio-temporal embedding loss is the weighted sum of the attraction, repulsion and regularisation losses:

$$\mathcal{L}_{instance} = \lambda_1 \mathcal{L}_a + \lambda_2 \mathcal{L}_r + \lambda_3 \mathcal{L}_{reg} \tag{4}$$

During inference, each pixel of the considered frame is assigned to an instance by randomly picking an unassigned point and aggregating close-by pixels with the mean-shift algorithm (Comaniciu & Meer, 2002). In the ideal case, with a test loss of zero, this will result in perfect clustering if the repulsion radius, $\rho_r$, is twice as large as the attraction radius, $\rho_a$.

## 3.1 SELF-SUPERVISED DEPTH ESTIMATION

The relative distance of objects is a strong cue to segment instances in space and time, as the motion of objects is temporally smooth. Knowing the previous distance of an object relative to the camera assists tracking as the future position will be constrained by the object's current location.

Depth estimation with supervised methods requires a vast quantity of high quality annotated data, which is challenging to acquire in a range of environments as laser measurements can be imprecise in natural scenes with motion and reflections. Because we have access to video in our instance segmentation dataset, we can leverage self-supervised depth losses from monocular videos, where the supervision comes from consecutive temporal frames. In addition to predicting the depth map, ego-motion also has to be inferred, but only during training to constrain the depth network.

Following Zhou et al. (2017) and Godard et al. (2019), we train a depth network with a separate pose estimation network with the hypothesis during training that scenes are mostly rigid, therefore assuming appearance change is mostly due to camera motion. The pixels that violate this assumption are masked from the view synthesis loss, as they otherwise create infinite holes during inference for objects that are typically seen in motion during training — more details in Appendix A.1. The training signal comes from novel view synthesis: generation of new images of the scene from a different camera pose. Let us denote by $(I_1, I_2, ..., I_T)$ a sequence of images, with target view $I_t$ and source view $I_s$. The view synthesis loss is given by:

$$\mathcal{L}_{vs} = \sum_{s \neq t} e(I_t, \hat{I}_{s \to t}) \tag{5}$$

with $\hat{I}_{s \to t}$ the synthesised view of $I_t$ from source image $I_s$ using the predicted depth $\hat{D}_t$ and the predicted $4 \times 4$ camera transformation $\hat{T}_{t \to s}$ predicted from the separate pose network. The projection error function, $e$, is described by Zhao et al. (2017) as a weighted sum of $L_1$, Structural Similarity Index (SSIM) and a smoothness regularisation term. Let us denote by $p_t$ the coordinate of a pixel in the target image $I_t$ in homogeneous coordinates. Given the camera intrinsic matrix, $K$, and the mapping $\varphi$ from image plane to camera coordinate, the corresponding pixel in the source image is provided by:

$$p_s \sim K\hat{T}_{t \to s}\varphi(K^{-1}p_t, \hat{D}_t(p_t)) \tag{6}$$

The projected coordinates $p_s$ are continuous values, we use the Spatial Transformer Network (Jaderberg et al., 2015) sampling mechanism to bilinearly interpolate the four neighbouring pixels to populate the reconstructed image $I_{s \to t}$.

Some pixels are visible in the target image, but are not in the source images, leading to a large projection error. As advocated by Godard et al. (2019), instead of summing, taking the minimum projection error greatly reduces artifacts due to occlusion and results in sharper predictions. The resulting view synthesis loss is:

$$\mathcal{L}_{vs} = \min_{s \neq t} e(I_t, \hat{I}_{s \to t}) \tag{7}$$

The resulting video instance embedding loss is the weighted sum of the attraction, repulsion, regularisation and geometric view synthesis losses:

$$\mathcal{L}_{\text{instance}} = \lambda_1 \mathcal{L}_a + \lambda_2 \mathcal{L}_r + \lambda_3 \mathcal{L}_{\text{reg}} + \lambda_4 \mathcal{L}_{\text{vs}} \tag{8}$$

## 4 MODEL ARCHITECTURE

Our model contains three components: an encoder, temporal model and the decoders. Each frame is first encoded to a more powerful and compact representation, then the temporal model learns the dynamics of the scene, and finally, the decoders output the instance embedding and depth prediction as illustrated by Figure 2.

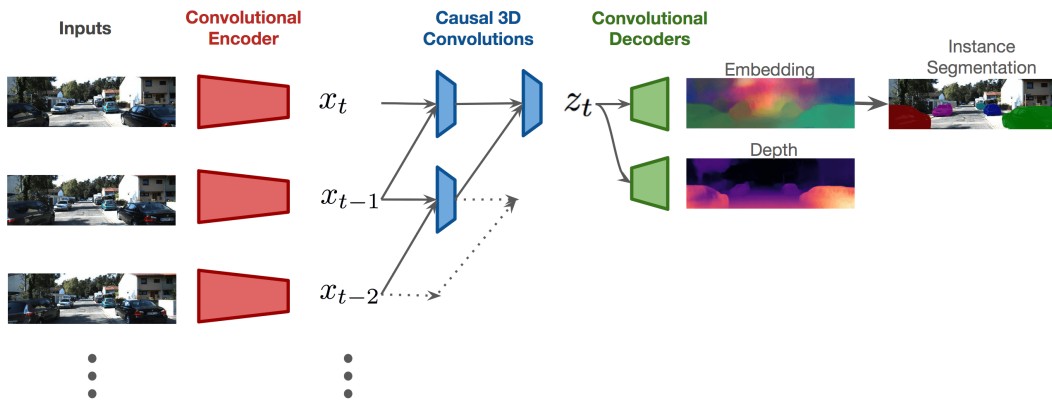

Figure 2: Our video instance segmentation and depth model architecture. The embedding, $z_t$, is trained to explicitly encode appearance, motion and geometry cues in order to predict an instance embedding and monocular depth prediction.

**Encoder.** We use a ResNet-18 (He et al., 2016) with 14.6M parameters as our encoder, which allows the network to run in real-time on sequences of images.

**Temporal Model.** The scene dynamics is learned with a causal 3D convolutional network composed of blocks of 3D residual convolutions (convolving in both space and time). For a given time index, $t$, the network only convolves over images from indices $s \leq t$ to compute the temporal representation $z_t$. It therefore does not use future frames and is completely causal. The temporal model does not decimate the spatial dimension of the encoding, but slowly accumulates information over time from the previous encodings $x_s$ with $s \leq t$. The temporal model is trained efficiently with convolutions as all input images are available during training, enabling parallel computations with GPUs. However, during inference, the model inherently has to be sequential, but can be made significantly faster by caching the convolutional features over time and eliminating redundant operations, as proposed by Paine et al. (2016) for WaveNet (Oord et al., 2016).

**Decoders.** The decoders then map the temporal encoding $z_t$ to its instance embedding $y_t$ of dimension $p \times height \times width$, with $p$ the embedding dimension, and depth $d_t$ of dimension $1 \times height \times width$. The embedding values belonging to the same instance are pushed together in the high-dimensional space $\mathbb{R}^p$, and pulled away from the other instances, over the whole video. Therefore, tracking instances simply requires comparing the mean embedding of a newly segmented instance with previously segmented instances. A distance lower than $\rho_r$ indicates a match.

To segment the instances, we first run a background mask network (trained separately) then we cluster the segmented embeddings using mean shift to discover dense regions of embeddings. Over time, the embeddings are accumulated up until the sequence length corresponding to the sequence length used during training to constrain instances spatio-temporally. This creates increasingly dense regions over time resulting in a better clustering. To ensure that embeddings of a particular instance can smoothly vary over time, the embeddings have a life span corresponding to the sequence length of the model.

**Pose and Mask Model.** For the pose network we use a ResNet and for the mask network we use an encoder-decoder model, also based on a ResNet. Further details are in Appendix A.1.

## 5  EXPERIMENTS

Next we describe experimental evidence which demonstrates the performance of our method by advancing the state-of-the-art on the KITTI Multi-Object and Tracking Dataset (Voigtlaender et al., 2019).

## 5.1 DATASET

The KITTI Multi-Object Tracking and Segmentation (MOTS) dataset (Voigtlaender et al., 2019) contains 8,008 frames with instance segmentation labels resulting in a total of 26,899 annotated cars. It is composed of 21 scenes with a resolution of $375 \times 1242$ with consistent instance ID labels across time, allowing the training of video instance segmentation models. The frames are annotated at 10 frames per second, which is suitable for self-supervised monocular depth prediction.

|  | Scenes | Frames | Annotations | Avg. # frames | Avg. # annotations |
|---|---|---|---|---|---|
| Train | 12 | 5,027 | 18,831 | 419 | 1,569 |
| Validation | 9 | 2,981 | 8,068 | 331 | 896 |

Table 1: Details of the KITTI Multi-Object Tracking and Segmentation (MOTS) dataset.

The ApolloScape dataset (Huang et al., 2018) also contains video instance segmentation labels for 49,287 frames, but the annotations are not consistent in time, rendering the training of a temporal model impossible. NuScenes (Caesar et al., 2019) features 1,000 scenes of 20 seconds with annotations at 2Hz in a diverse range of environments (different weather, daytime, city) but only contains bounding box labels, failing to represent the fine-grained details of instance segmentation. Temporal instance segmentation is also available on short snippets of the DAVIS dataset (Pont-Tuset et al., 2017), but each snippet is recorded by a different camera and is too short to effectively learn a depth model. For this reason, we focus on the KITTI MOTS dataset – it is the only dataset that contains consistent video instance segmentation in a sufficient quantity to train deep models.

## 5.2 HYPER-PARAMETERS

We halve the input images to our encoder to use an input RGB resolution of $3 \times 192 \times 640$. The resulting encoding is $128 \times 24 \times 80$. The decoders then map the temporal encoding $z_t$ to its instance embedding $y_t$ of dimension $p \times 192 \times 640$, with $p = 8$ the embedding dimension, and depth $d_t$ of dimension $1 \times 192 \times 640$. Except for the experiments in Table 4, we train with a sequence length of 5 which corresponds to 0.5 seconds of temporal context since the videos are 10Hz.

In the loss function, we set the attraction radius $\rho_a = 0.5$ and repulsion radius $\rho_r = 1.5$. We weight the losses with attraction and repulsion loss weight $\lambda_1 = \lambda_2 = 1.0$, regularisation loss $\lambda_3 = 0.001$ and depth loss $\lambda_4 = 1.0$.

## 5.3 METRICS

Let us define multi-object tracking and segmentation metrics, which measures the quality of the segmentation as well as the consistency of the predictions over time. Contrary to bounding box detection, where a ground truth box may overlap with several predicted boxes, in instance segmentation, since each pixel is assigned to at most one instance, only one predicted mask can have an Intersection over Union (IoU) larger than a given threshold with a given ground truth mask. Let us denote by $\mathbb{H}$ the set of predicted ids, $\mathbb{M}$ the set of ground truth ids and $g$ the mapping from hypothesis masks to ground truth masks. $g : \mathbb{H} \to \mathbb{M} \cup \emptyset$ is defined as:

$$g(h) = \begin{cases} \text{argmax}_m \text{IoU}(h, m), & \text{if } \max_m \text{IoU}(h, m) > \text{threshold} \\ \emptyset, & \text{otherwise} \end{cases} \qquad (9)$$

We thus define:

- True positives as: $TP = \{h \in \mathbb{H} | g(h) \neq \emptyset\}$, correctly assigned predicted masks.
- False positives as: $FP = \{h \in \mathbb{H} | g(h) = \emptyset\}$, predicted masks not assigned to any ground truth mask.
- False negatives as: $FN = \{m \in \mathbb{M} | g^{-1}(m) = \emptyset\}$, ground truth masks not covered by any hypothesis mask.
- Soft number of true positives: $\tilde{TP} = \sum_{h \in TP} \text{IoU}(h, g(h))$

Let the function $pred : \mathbb{M} \rightarrow \mathbb{M} \cup \emptyset$ map a ground truth mask to its latest tracked predecessor ($\emptyset$ if the ground truth mask is seen for the first time). The set $IDS$ of ID switches is defined as the set of ground truth masks whose predecessor was tracked by a different ID.

Following Voigtlaender et al. (2019), we define the following MOTS metrics: multi-object tracking and segmentation precision (MOTSP), multi-object tracking and segmentation accuracy (MOTSA) and finally the soft multi-object tracking and segmentation accuracy (sMOTSA) that measures segmentation as well as detection and tracking quality.

$$\text{MOTSP} = \frac{|\tilde{TP}|}{TP} \tag{10}$$

$$\text{MOTSA} = 1 - \frac{|FP| + |FN| + |IDS|}{|\mathbb{M}|} = \frac{|TP| - |FP| - |IDS|}{|\mathbb{M}|} \tag{11}$$

$$\text{sMOTSA} = \frac{|\tilde{TP}| - |FP| - |IDS|}{|\mathbb{M}|} \tag{12}$$

We also measure the average precision (AP), i.e. the normalised area under the precision/recall curve.

## 5.4 RESULTS

We compare our model to the following baselines for video instance segmentation and report the results in Table 2.

- **Single-frame embedding loss** (Brabandere et al., 2017), previous state-of-the-art method where instance segmentations are propagated in time using intersection-over-union association.
- **Without temporal model**, spatio-temporal embedding loss, without the temporal model.
- **Without depth**, temporal model and spatio-temporal embedding loss, without the depth loss.

We also report the results of Mask R-CNN (He et al., 2017) and Track R-CNN (Voigtlaender et al., 2019), even though direct comparison with the latter is not possible as their model was pretrained on Cityscapes and Mapillary Vistas, and is not causal as they use future frames to predict the current frame instance segmentations.

| | KITTI only | Causal | MOTSA | sMOTSA | MOTSP | AP | recall | precision |
|---|---|---|---|---|---|---|---|---|
| Brabandere et al. (2017) | ✓ | ✓ | 0.575 | 0.423 | 0.803 | 0.612 | 0.770 | 0.841 |
| Mask R-CNN | ✗ | ✓ | 0.584 | 0.455 | **0.833** | **0.646** | 0.771 | **0.875** |
| Voigtlaender et al. (2019) | ✗ | ✗ | 0.671 | 0.540 | 0.831 | 0.656 | 0.775 | 0.894 |
| Without temporal model | ✓ | ✓ | 0.582 | 0.426 | 0.799 | 0.607 | 0.777 | 0.835 |
| Without depth | ✓ | ✓ | 0.591 | 0.433 | 0.801 | 0.614 | **0.795** | 0.822 |
| **Ours** | ✓ | ✓ | **0.613** | **0.461** | 0.801 | 0.600 | 0.764 | 0.839 |

Table 2: KITTI MOTS validation set results comparing our model with baseline approaches.

The static detection metrics (average precision, recall, precision) are evaluated image by image without taking into account the temporal consistency of instance segmentations. As the compared models (Without temporal model, Without depth, Ours) are all using the same mask network, they show similar performance in terms of detection. However, when evaluating performance on metrics that measures temporal consistency (MOTSA and sMOTSA), our best model shows significant improvement over the baselines.

The variant without the temporal model performs poorly as it does not have any temporal context to learn a spatio-temporal embedding and therefore only relies on spatial appearance. The temporal model on the other hand learns with the temporal context and local motion, which results in a better embedding. Our model, which learns to predict both a spatio-temporal embedding and monocular depth, achieves the best performance. In addition to using cues from appearance and temporal

context, estimating depth allows the network to use information from the relative distance of objects to disambiguate them. Finally, we observe that our model outperforms Mask R-CNN (He et al., 2017) on the temporal metrics (MOTSA and sMOTSA) even though the latter exhibits a higher detection accuracy, further demonstrating the temporal consistency quality of our spatio-temporal embedding.

### 5.4.1 ANALYSIS OF CLUSTERING AND MASK SEGMENTATION.

Our model first relies on the mask segmentation to isolate which pixel locations to consider for instance clustering. We evaluate the impact of using the ground truth mask against our predicted mask in Table 3. The performance gain is significant, hinting that a better instance segmentation would be possible by improving the mask network.

Next, we evaluate the effect of clustering. In the best scenario, the validation loss would be zero, and the clustering would be perfect using the MeanShift algorithm. However, this scenario is unlikely and the clustering algorithm is affected by noisy embeddings. We evaluate the effect of this noise by clustering with the ground-truth mean for each instance, by thresholding with $\rho_r$ around the ground truth instance embedding mean. This also results in a boost in the evaluation metrics, but most interestingly, a model that uses both ground truth instance embedding mean clustering and ground truth mask performs worse than a model segmented with ground truth mask and our clustering algorithm. This is because our clustering algorithm accumulates embeddings from past frames and therefore creates an attraction force for the mean shift algorithm that enables the instances to be matched more consistently.

| GT Mean | GT Mask | MOTSA | sMOTSA | MOTSP | AP | Recall | Precision |
|---------|---------|-------|--------|-------|-----|--------|-----------|
| ✗ | ✗ | 0.613 | 0.461 | 0.801 | 0.600 | 0.764 | 0.839 |
| ✓ | ✗ | 0.700 | 0.616 | 0.897 | 0.691 | 0.816 | 0.886 |
| ✗ | ✓ | **0.804** | **0.751** | **0.936** | **0.786** | **0.832** | **0.971** |
| ✓ | ✓ | 0.714 | 0.644 | 0.915 | 0.710 | 0.824 | 0.892 |

Table 3: Comparing the effect of noisy against ground-truth clustering and mask segmentation on the KITTI MOTS dataset.

### 5.4.2 EFFECT OF THE SEQUENCE LENGTH

Our model learns a spatio-temporal embedding that clusters video-pixels from a given instance. Correspondence of instances between frames is achieved by matching detected instances to previous instances if the embedding distance is below the repulsion radius, $p_a$. Instance tracking can occur for an arbitrarily long sequence of time, as long as the embedding changes smoothly over time, which is likely the case as temporal context and depth must evolve gradually. However, when the spatio-temporal embedding is trained over sequences which are too long, the embedding learning collapses. This is because the attractive loss term is detrimental between distant frames, it pressures pixels from the same instance to be have corresponding embeddings when their appearance and depth is no longer similar. It also suggests our model is able to reason over lower order motion cues more effectively than longer term dynamics. This is seen experimentally in Table 4.

| Length | 1 | 3 | 5 | 7 | 10 | 15 |
|--------|-----|-----|-----|-----|-----|-----|
| MOTSA | 0.575 | 0.590 | **0.613** | 0.555 | 0.538 | 0.402 |
| sMOTSA | 0.423 | 0.435 | **0.461** | 0.402 | 0.398 | 0.273 |
| MOTSP | 0.803 | **0.810** | 0.801 | 0.788 | 0.792 | 0.783 |

Table 4: Influence of the sequence length on model performance. This indicates that our model can learn short-term motion features effectively, but not long-term cues. We reason that this is because over longer sequences, the loss prevents the embedding smoothly shifting, which naturally occurs to changing pose, appearance, context and lighting in the scene. We find the optimum sequence length on this dataset to be five.

## 5.5 QUALITATIVE EXAMPLES

The instance segmentation of our model is consistent across frames as instances are clustered in both space and time. This provides more robust clustering compared to a per-frame approaches. We demonstrate this with the following scenarios showing tracking through partial (Figure 3 and full occlusion (Figure 5), as well as continuous tracking through noisy detections (Figure 4). Additional examples and failure cases of our model are shown in Appendix A.2 and a video demo can be viewed here: https://youtu.be/pqRPXRUlQ2I

In each example, we show from left to right: RGB input image, ground truth instance segmentation, predicted instance segmentation, embedding visualised in 2D, embedding visualised in RGB and predicted monocular depth. The embedding is visualised in 2D with the corresponding mean shift clustering. Each color represents a different instance, the inner circle is the attraction radius of the instance mean embedding, and the outer circle is the repulsion radius of each instance. Additionally, we also visualise the embedding spatially in 3D, by projecting its three principal components to an RGB image.

We show in Appendix A.2 that incorporating depth context greatly improves the quality of the embedding, especially in complex scenarios such as partial or total occlusion. We also observe that the embedding is much more structured when incorporating 3D geometry information.

## 6 CONCLUSIONS

We proposed a new spatio-temporal embedding loss that generates consistent instance segmentation over time. The temporal network models the past temporal context and the depth network constrains the embedding to aid disambiguation between objects. We demonstrated that our model could effectively track occluded instances or instances with missed detections, by leveraging the temporal context. Our method advanced the state-of-the-art at video instance segmentation on the KITTI Multi-Object and Tracking Dataset.

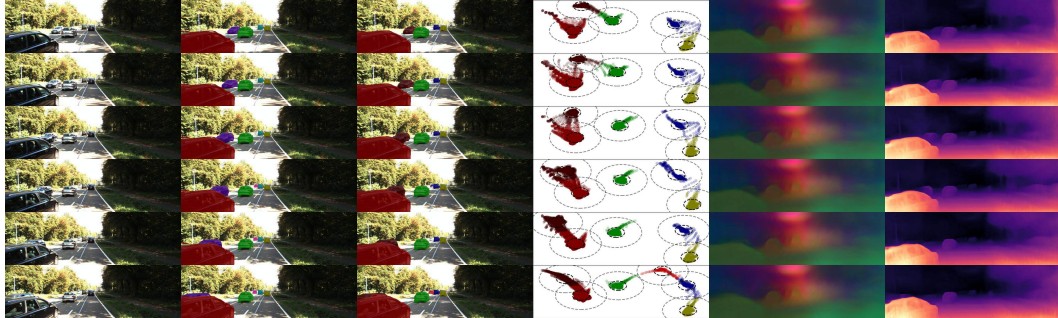

Figure 3: **Partial occlusion.** The segmented brown car is correctly segmented even when being partially occluded by the segmented red car, as the embedding contains past temporal context and is aware of the motion of brown car.

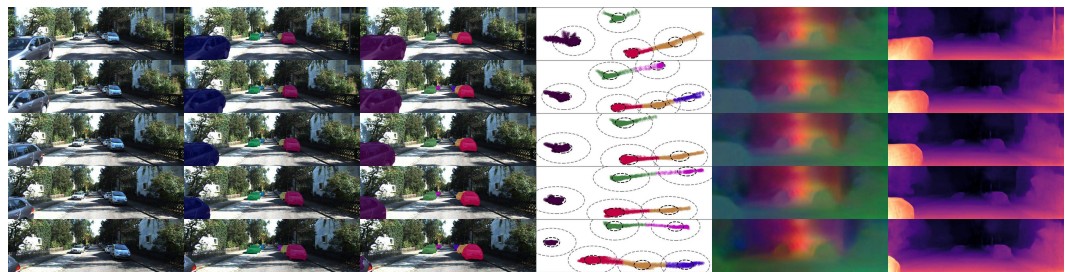

Figure 4: **Continuous tracking.** The segmented pink and purple cars are accurately tracked even with missing detections.

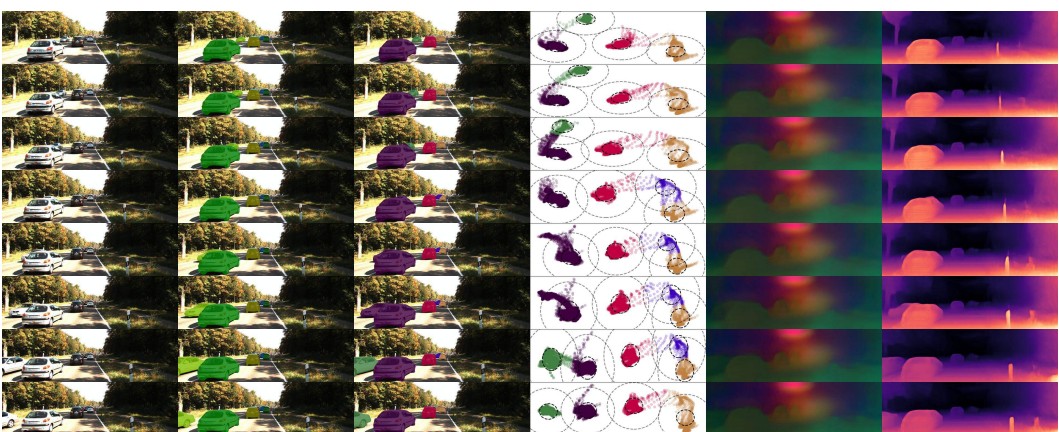

Figure 5: **Total occlusion.** The segmented green car correctly tracked, even though it was completely occluded by another car.

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

# A APPENDIX

## A.1 NETWORK ARCHITECTURE DETAILS

We report the details of each component of our model in this section. The number of parameters and layers of each module are in Table 5.

|            | Encoder | Temporal | Decoders | Pose  | Mask  |
|------------|---------|----------|----------|-------|-------|
| Parameters | 14.6M   | 0.7M     | 0.4M     | 13.0M | 14.8M |
| Layers     | 18      | 36       | 7        | 22    | 25    |

Table 5: Number of parameters and layers of each module.

**Encoder.** The encoder is a ResNet-18 convolutional layer (He et al., 2016), with 128 output channels.

**Temporal model.** The temporal model contains 12 residual 3D convolutional blocks, with only the first and last block convolving over time. Each residual block is the succession of: projection layer of kernel size $1 \times 1 \times 1$ to halve the number of channels, 3D causal convolutional layer $t \times 3 \times 3$, projection layer $1 \times 1 \times 1$ to double the number of channels.

We set the temporal kernel size to $t = 2$, and the number of output channels to 128.

**Decoders.** The decoders for instance embedding and depth estimation are identical and consist of 7 convolutional layers with channels [64, 64, 32, 32, 16, 16] and 3 upsampling layers. The final convolutional layer contains $p$ channels for instance embedding and 1 channel for depth.

**Depth Masking.** During training, we remove from the photometric reprojection loss the pixels that violate the rigid scene assumption, i.e. the pixels whose appearance do not change between adjacents frames. We set the mask $M$ to only include pixels where the reprojection error is lower with the warped image $\hat{I}_{s \to t}$ than the unwarped source image $I_s$:

$$M = \left[ \min_s e(I_t, \hat{I}_{s \to t}) < \min_s e(I_t, I_s) \right]$$

**Pose Network.** The pose network is the succession of a ResNet-18 model followed by 4 convolutions with [256, 256, 256, 6] channels. The last feature map is averaged to output a single 6-DoF transformation matrix.

**Mask Network.** The mask network is trained separately to mask the background and is the succession of the Encoder and Decoder described above.

## A.2 ADDITIONAL QUALITATIVE EXAMPLES

The following examples show qualitative results and failure examples of our video instance segmentation model on the KITTI Multi-Object and Tracking Dataset. From left to right: RGB input image, ground truth instance segmentation, predicted instance segmentation, embedding visualised in 2D, embedding visualised in RGB and predicted monocular depth.

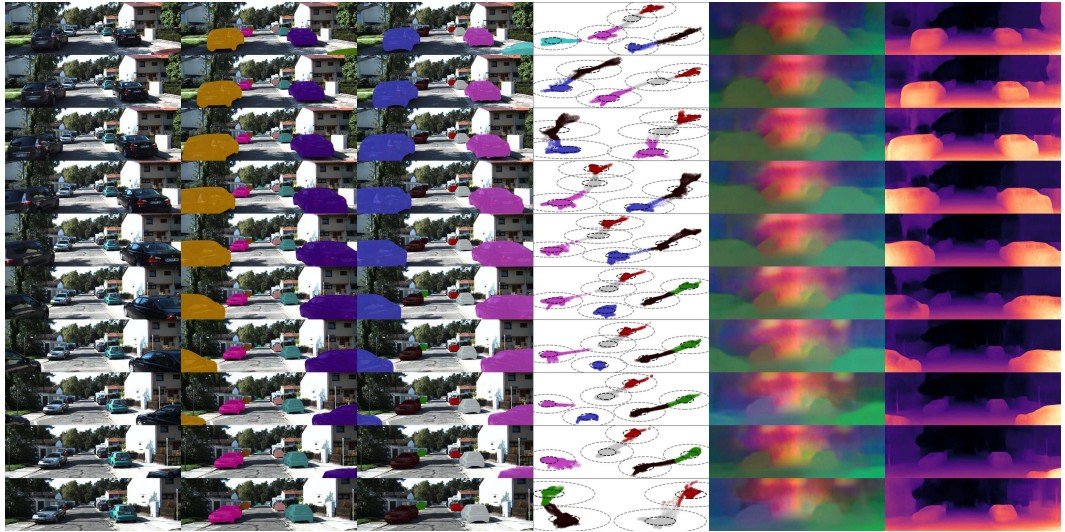

Figure 6: Video instance segmentation of parked cars.

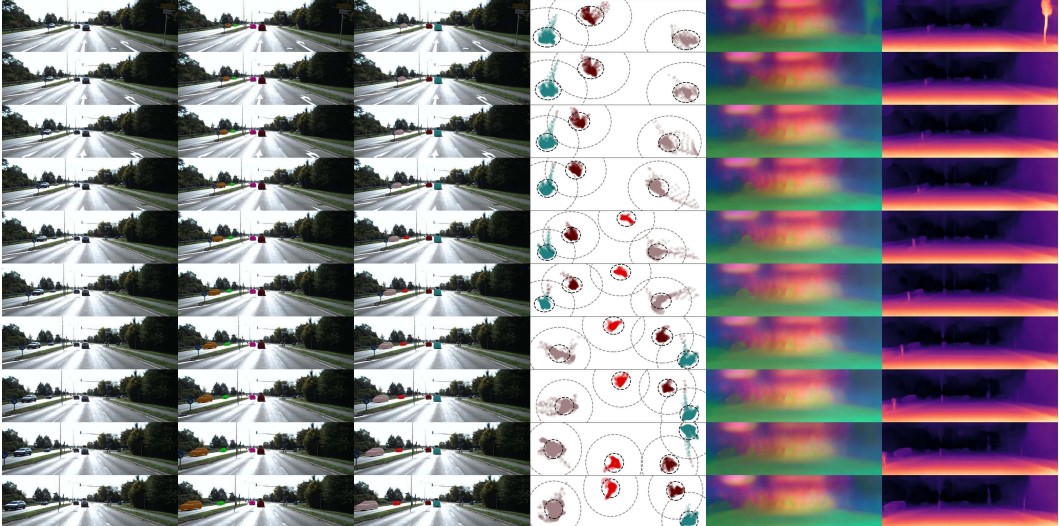

Figure 7: Video instance segmentation of other traffic.

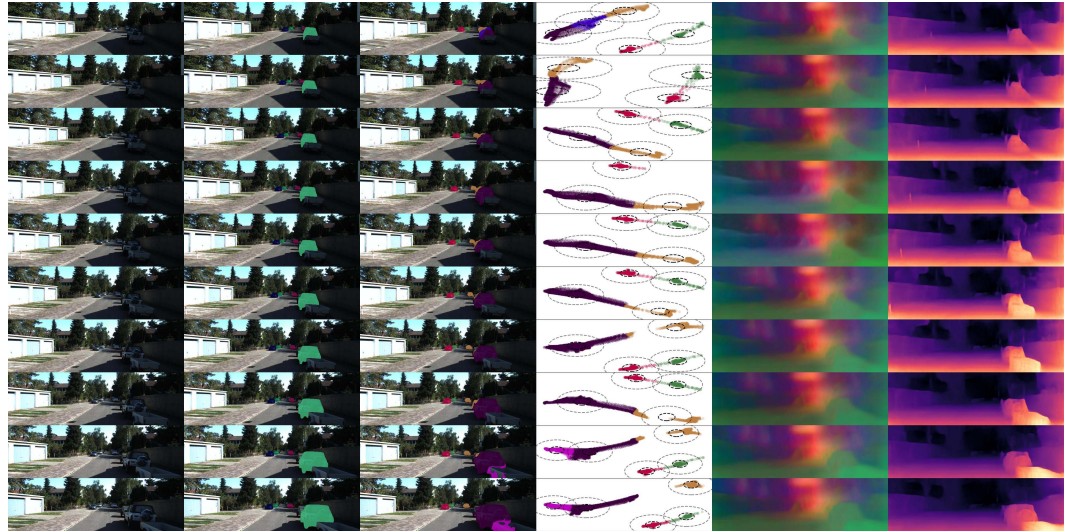

Figure 8: Failure case: the vehicle is segmented into two separate instances.

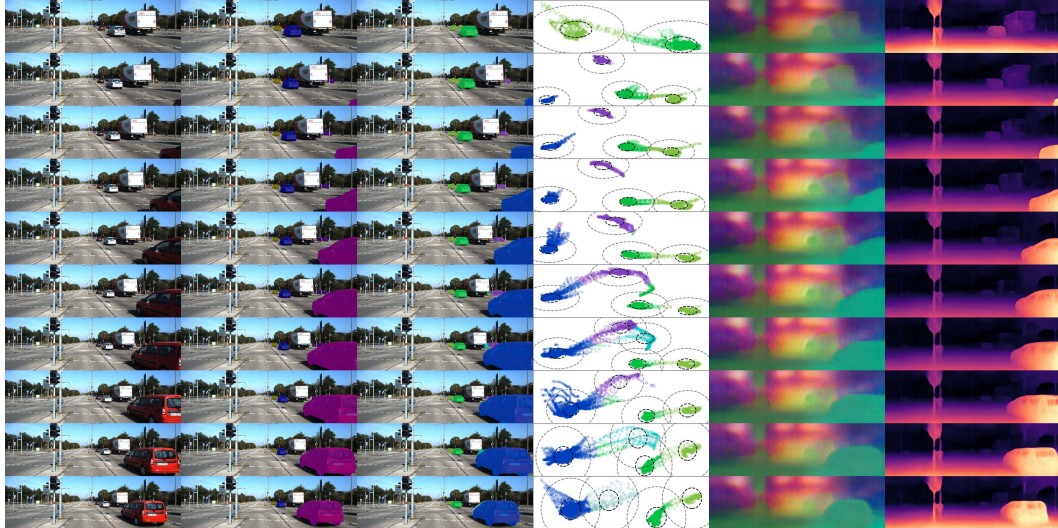

Figure 9: Failure case: two far-away cars are segmented as one instance.

### A.3 QUALITATIVE IMPROVEMENTS WITH DEPTH

We show that our model greatly benefits from depth estimation, with the learned embedding being more structured, and correctly tracking objects in difficult scenarios such as partial or total occlusion.

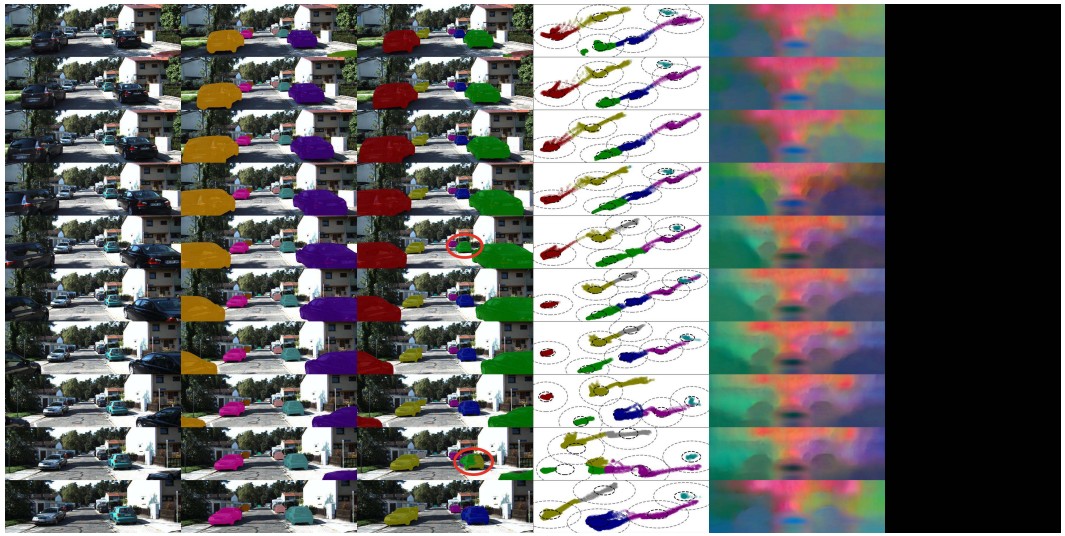

(a) Without depth estimation.

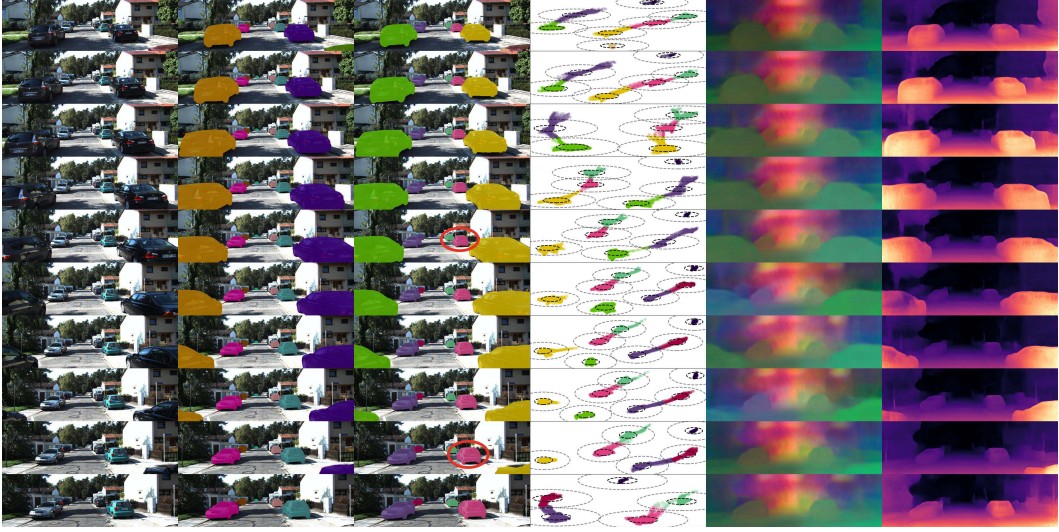

(b) With depth estimation

Figure 10: Without depth, the car circled in red is wrongly tracked in frame 5 and 9, while our model correctly tracks it as the network has learned a consistent embedding based not only on appearance, but also on 3D geometry. Also, the RGB projection of the embedding from our model is considerably better and much more structured.

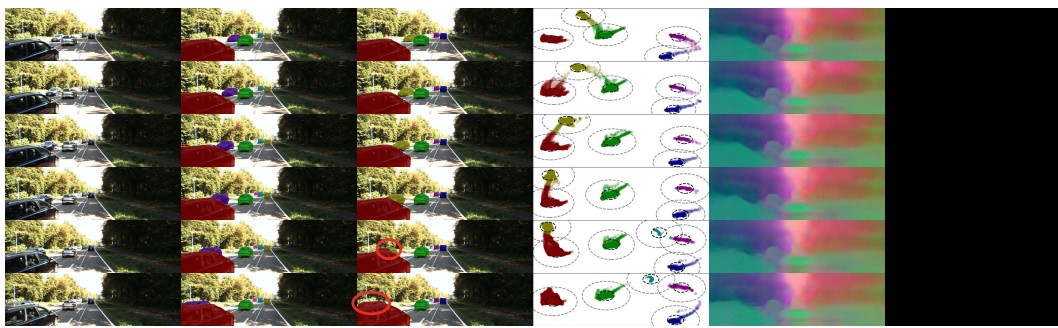

(a) Without depth estimation.

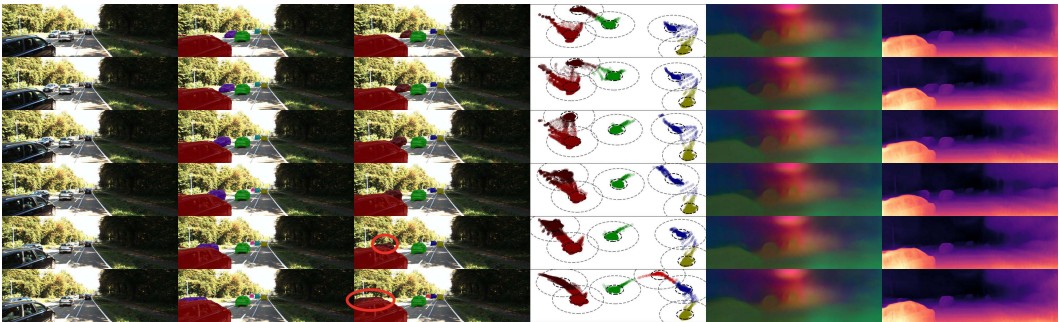

(b) With depth estimation

Figure 11: Without depth, the circled car merges into the red-segmented car, while our model does not as there is a significant difference in depth between the two cars.

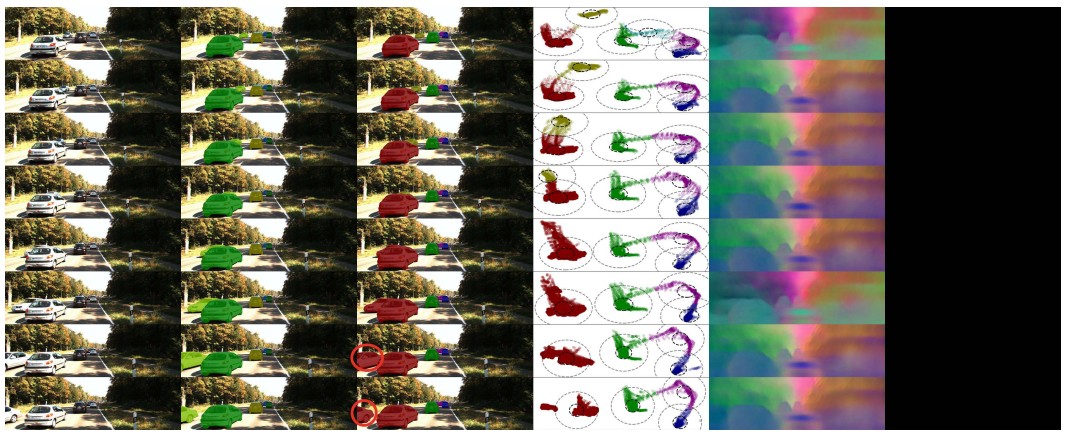

(a) Without depth estimation.

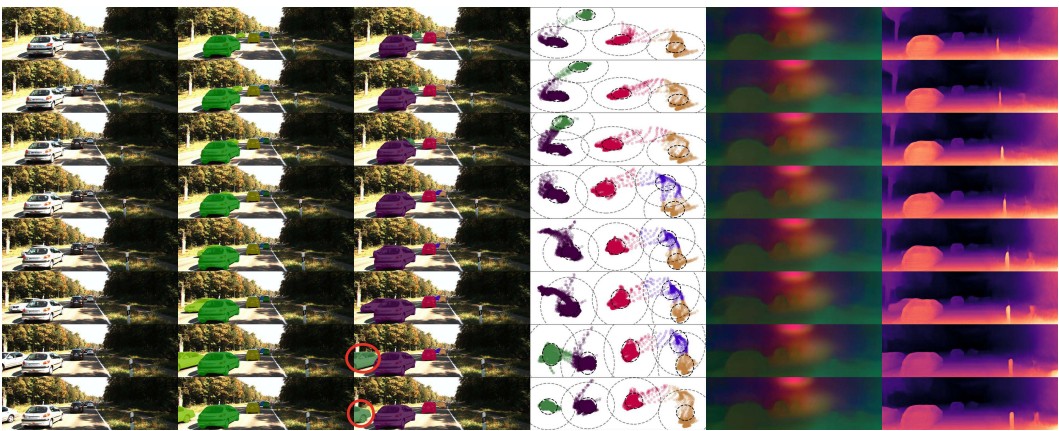

(b) With depth estimation

Figure 12: The model without depth is not able to handle complete occlusion, while ours can.

