# OpenReview forum: "Learning a Spatio-Temporal Embedding for Video Instance Segmentation"
_ICLR.cc/2020/Conference — Reject_

### Official Review · AnonReviewer2 · 2019-10-20
**Official Blind Review #2**

**Rating:** 3

**Review:**

Summary
The paper presents a method to learn an embedding space for each pixel in a video that indicates the instance id of the objects. They also propose an auxiliary loss based on depth prediction to improve performance. This can be used to both segment and track objects in videos.

Strengths
1) The proposed approach is simple and general and handles the problem of occluding objects in videos.
2) Their causal convolution architecture will be useful for other problems in videos.
3) The authors perform a number of ablations to investigate how much each part of their solution contributes to the final performance.
4) The paper is well-written and well-motivated.

Weaknesses
1) Comparisons to other video instance segmentation methods [1,2,3] are missing. The only comparison is done with a single-frame instance embedding method.
2) The authors propose to predict depth as an auxiliary task. However, they do not use the predicted depth at test time. This is a missed opportunity. Difference in depth might help in identifying instances. Also, it might be worthwhile to investigate if instance segmentation is helping depth prediction.
3) Experiments have been conducted on only one dataset.


References
[1] "Video Instance Segmentation" Linjie Yang and Yuchen Fan.
[2] "MaskRNN: Instance Level Video Object Segmentation" Yuan-Ting Hu, Jia-Bin Huang, and Alexander G. Schwing.
[3] "SAIL-VOS: Semantic Amodal Instance Level Video Object Segmentation – A Synthetic Dataset and Baselines" Yuan-Ting Hu, Hong-Shuo Chen, Kexin Hui, Jia-Bin Huang, Alexander Schwing.



**Experience Assessment:**

I have published one or two papers in this area.

**Review Assessment: Checking Correctness Of Derivations And Theory:**

N/A

**Review Assessment: Checking Correctness Of Experiments:**

I carefully checked the experiments.

**Review Assessment: Thoroughness In Paper Reading:**

I read the paper thoroughly.

---

> ### Author Response · Authors · 2019-11-13
> **Response to Reviewer2**
>
> Thank you for your helpful comments and suggestions. Here’s our answer to the concerns you have raised:
>
> 1. "Comparisons to other video instance segmentation methods [1,2,3] are missing. The only comparison is done with a single-frame instance embedding method."
>
> We have updated the paper to include the results of Track R-CNN [4], more details in the general response (point 1). The approach of Yang and Fan [1] is almost identical to Track R-CNN, as they also modify Mask R-CNN to include a Tracking head which assigns an identity vector to each detection. The implementation of Hu et al. [2] is not described in the paper, and there is no publicly available code associated. I have contacted the authors to ask for clarifications in the implementation of their model but they unfortunately haven’t responded yet. Hu et al. [3] introduced a new synthetic dataset for Video Object Segmentation, but did not propose a new model as they give benchmark results using Mask R-CNN. We have however added a comparison with Mask R-CNN and IoU correspondence to track instances.
>
> We also updated our Related Work section to include [2] and [3].
>
> 2. "The authors propose to predict depth as an auxiliary task. However, they do not use the predicted depth at test time. This is a missed opportunity. Difference in depth might help in identifying instances. Also, it might be worthwhile to investigate if instance segmentation is helping depth prediction."
>
> We have actually ran this experiment. In order to more explicitly use depth information, we concatenated the predicted depth map to the segmentation embedding and learned a new embedding from these features. However, the results did not differ from our model, as the shared representation (after the 3D Causal convolutions) between embedding and depth already encodes information from 3D geometry.
>
> We agree that depth prediction and instance segmentation might benefit from each other, and it is one of our future research directions.
>
> References
> [1] "Video Instance Segmentation" Linjie Yang and Yuchen Fan.
> [2] "MaskRNN: Instance Level Video Object Segmentation" Yuan-Ting Hu, Jia-Bin Huang, and Alexander G. Schwing.
> [3] "SAIL-VOS: Semantic Amodal Instance Level Video Object Segmentation – A Synthetic Dataset and Baselines" Yuan-Ting Hu, Hong-Shuo Chen, Kexin Hui, Jia-Bin Huang, Alexander Schwing.
> [4] “MOTS: Multi-Object Tracking and Segmentation” Paul Voigtlaender, Michael Krause, Aljosa Osep, Jonathon Luiten, Berin Balachandar Gnana Sekar, Andreas Geiger, Bastian Leibe.

---

### Official Review · AnonReviewer3 · 2019-10-23
**Official Blind Review #3**

**Rating:** 3

**Review:**

This paper propose a video instance embedding loss for jointly tackling the instance tracking and depth estimation from self-supervised learning.

Pros:
1: I think the method is moving towards the right direction that 3d geometry and 2d instance representation should be considered jointly under the scenario of video learning.
2: The video instance embedding loss is also making sense as an extension of image instance embedding.

Cons:
1: I think the major argument I have is this method is lack of technical novelty, since it is straight forward to adopt the loss of  Brabandere et.al 2017 to video cases for including pixels in the same group under ground truth tracking, and the self-supervised loss is exactly the same as previous methods. The fusion between depth and segments are relatively weak since it just ask the embedding to also decode depth, is there any further analysis of visual effect of explaining where the depth helps segments?

2: In the experiments, the baseline for comparison over MOTS is fairly old, and I think it makes sense to include the number of MOTS paper, which is currently hard to align with that shown in the paper.  In Tab.2, the author only highlight the improved motion metric, while in per-frame AP the results are actually lower than the baselines. It also needs to be well explained.

3: The paper claims `"it generates temporal consistent segmentation " (which is not guaranteed, maybe just statistically better but not exact).

Overall, in my opinion I suggest it to be a workshop paper, but the contribution is somehow not significant for a major publication.


**Experience Assessment:**

I have published in this field for several years.

**Review Assessment: Checking Correctness Of Derivations And Theory:**

I carefully checked the derivations and theory.

**Review Assessment: Checking Correctness Of Experiments:**

I assessed the sensibility of the experiments.

**Review Assessment: Thoroughness In Paper Reading:**

I read the paper thoroughly.

---

> ### Author Response · Authors · 2019-11-13
> **Response to Reviewer3**
>
> Many thanks for your careful review and helpful comments. Here’s our answer to the concerns you have raised:
>
> 1a. "I think the major argument I have is this method is lack of technical novelty, since it is straight forward to adopt the loss of  Brabandere et.al 2017 to video cases for including pixels in the same group under ground truth tracking, and the self-supervised loss is exactly the same as previous methods."
>
> Although it might be straightforward to extend the loss of Brabandere et al. [1] to time, it is challenging to design an architecture that jointly integrates context from motion (3D Causal convolutions) and geometry (self-supervised depth estimation) to learn a spatio-temporal embedding that can consistently segment instances over time.
>
> 1b. "The fusion between depth and segments are relatively weak since it just ask the embedding to also decode depth, is there any further analysis of visual effect of explaining where the depth helps segments?"
>
> Incorporating depth context greatly improves the quality of the embedding as shown in the three examples added in the Appendix (section A.3).
> We compare the outputs of the model trained with and without self-supervised depth estimation. For each figure, we have from left to right: RGB image, ground truth segmentation, predicted segmentation, embedding visualised in 2D, embedding visualised in RGB by projecting the three main components in the image space, and depth map.
>
> (i) Without depth, the car circled in red is wrongly tracked in frame 5 and 9, while our model correctly tracks it as the network has learned a consistent embedding based not only on appearance, but also on 3D geometry. Also, the RGB projection of the embedding from our model is considerably better and much more structured.
> (ii) Without depth, the circled car merges into the red-segmented car, while our model does not as there is a significant difference in depth between the two cars.
> (iii) The model without depth is not able to handle complete occlusion, while ours can.
>
> 2. "In the experiments, the baseline for comparison over MOTS is fairly old, and I think it makes sense to include the number of MOTS paper, which is currently hard to align with that shown in the paper. In Tab.2, the author only highlight the improved motion metric, while in per-frame AP the results are actually lower than the baselines. It also needs to be well explained."
>
> Please refer to the general response (point 1 and 3).
>
> 3. "The paper claims `"it generates temporal consistent segmentation " (which is not guaranteed, maybe just statistically better but not exact)."
>
> Quantitatively, we show that our model improves the baselines with IoU correspondence, and qualitatively we can see that the segmentation is temporally consistent as shown in the accompanying video: https://youtu.be/pqRPXRUlQ2I
> And on more qualitative video examples here: https://drive.google.com/open?id=1u-kGxQEWIoC6FguUiXFHyxUcOiG2iIIf
>
>
> References
> [1] “Semantic Instance Segmentation with a Discriminative Loss Function” Bert De Brabandere, Davy Neven, Luc Van Gool.

---

### Official Review · AnonReviewer1 · 2019-10-23
**Official Blind Review #1**

**Rating:** 6

**Review:**

This paper presents learning a spatio-temporal embedding for video instance segmentation. With spatio-temporal embedding loss, it is claimed to generate temporally consistent video instance segmentation. The authors show that the proposed method performs nicely on tracking and segmentation task, even when there are occlusions.

Overall, this paper is well-written. Section 3 clearly explains the loss functions. The main idea is not very complex, but generally makes sense. The authors mention that scenes are assumed to be mostly rigid, and appearance change is mostly due to the camera motion. I would like to see more argument about this, as there are cases if this is obviously not true; for instance, human changes pose significantly. If we limit the range of discussion to some narrow domain, such as self-driving, this might be more valid, but we may want to see some discussion about validity of this assumption.

Some modules are not full explained in detail. For example, what is the background mask network? Which model was used, and how was it trained?

In experiment, the proposed method shows nice score on MOTSA and sMOTSA, but all other metrics, it is on the worse side. The authors are encouraged to discuss more about the metrics and experimental results with the other metrics as well. Other than these, the experiment was well-designed and conducted.

**Experience Assessment:**

I do not know much about this area.

**Review Assessment: Checking Correctness Of Derivations And Theory:**

N/A

**Review Assessment: Checking Correctness Of Experiments:**

I assessed the sensibility of the experiments.

**Review Assessment: Thoroughness In Paper Reading:**

I made a quick assessment of this paper.

---

> ### Author Response · Authors · 2019-11-13
> **Response to Reviewer1**
>
> Many thanks for the feedback. Here’s our answer to the concerns you have raised:
>
> 1. “The authors mention that scenes are assumed to be mostly rigid, and appearance change is mostly due to the camera motion. I would like to see more argument about this, as there are cases if this is obviously not true; for instance, human changes pose significantly. If we limit the range of discussion to some narrow domain, such as self-driving, this might be more valid, but we may want to see some discussion about validity of this assumption.”
>
> Our model learns depth in a self-supervised way using a photometric reconstruction loss, which operates under the assumption of a moving camera and a static scene. When this hypothesis does not hold true, for example when the camera is stationary or some scene objects are in motion, performance can rapidly deteriorate. During test time, objects that are typically seen in motion during training are then assigned an infinite depth value. To overcome this problem, we simply mask during training the pixels that do not change appearance from frame to frame (details in Appendix A.1). Since these pixels are often associated with objects moving at the same velocity as the camera, or to scenarios when the camera stops moving, this masking approach effectively removes the pixels that violates the rigid scene assumption.
>
> During inference however, our model is able to correctly predict the depth maps of moving objects: see some examples here https://drive.google.com/open?id=1u-kGxQEWIoC6FguUiXFHyxUcOiG2iIIf
>
> 2. "Some modules are not full explained in detail. For example, what is the background mask network? Which model was used, and how was it trained?"
>
> The background mask network is described in Appendix A.1: it is a ResNet network with a U-net structure and was trained on KITTI.
>
> 3. "In experiment, the proposed method shows nice score on MOTSA and sMOTSA, but all other metrics, it is on the worse side. The authors are encouraged to discuss more about the metrics and experimental results with the other metrics as well."
>
> Please refer to the general response (point 3).

---

### Author Response · Authors · 2019-11-13
**General response to the reviewers**

We would like to thank the reviewers for their feedback and helpful comments.

We wanted to emphasise that our work presents the first spatio-temporal embedding approach for Video Instance Segmentation. All the other existing methods (Hu et al. [1], Voigtlaender et al. [2], Yang and Fan [3]) follow the region proposal approach, i.e. region of interest detection followed by mask refinement and identification vector assignment to track objects. We believe this is clear demonstration of novelty that is interesting to the ICLR community.

We propose a different paradigm more grounded with the real-world:
(i) Our method learns a spatio-temporal embedding integrating cues from appearance, motion and 3D geometry, which can naturally track instances over time, without any complex postprocessing.
(ii) Our network runs in real-time and online as our architecture is entirely causal ‒ we do not incorporate information from future frames as opposed to previous methods.


In addition to addressing individual reviewer concerns, we have revised the paper and would like to highlight these improvements:

1. We report the results of the approach used by Hu et al. [1], and Voigtlaender et al. (Track R-CNN [2]) using the implementation of the authors. We observe that our model is competitive even though a direct comparison with Track R-CNN is not possible as:
(i) Their model was pretrained on Cityscapes and Mapillary Vistas while our model was solely trained on KITTI.
(ii) Track R-CNN operates on future frames to predict the current segmentation, while our model is causal and only uses past and present frames.

It is possible to further improve our model by using a more powerful mask network, as the quality of the mask has a great influence on performance: for example when using the ground truth mask, our MOTSA metric goes from 0.612 to 0.804.

2. We added qualitative examples that illustrates cases where learning 3D geometry is essential to disambiguate between objects, especially in complex scenarios such as partial or total occlusion (see the figures in Appendix A.3). We also observe that the embedding is much more structured when incorporating depth information.

3. The static detection metrics (average precision, recall, precision) are evaluated image by image without taking into account the temporal consistency of instance segmentations. As the compared models (Without temporal model, Without depth, Ours) are all using the same mask network, they show similar performance in terms of detection.

However, when evaluating performance on metrics that measure temporal consistency (MOTSA and sMOTSA), our best model shows significant improvement over the baselines.

4. We updated our Related Work section to include the work of [1] and [4] on Video Object Segmentation.

5. We clarified how our model learns depth by specifying that during training we mask pixels that violate the rigid scene assumption in the photometric reconstruction loss (more details in Appendix A.1).

All modifications are in green in the paper.


References
[1] "SAIL-VOS: Semantic Amodal Instance Level Video Object Segmentation – A Synthetic Dataset and Baselines" Yuan-Ting Hu, Hong-Shuo Chen, Kexin Hui, Jia-Bin Huang, Alexander Schwing.
[2] “MOTS: Multi-Object Tracking and Segmentation” Paul Voigtlaender, Michael Krause, Aljosa Osep, Jonathon Luiten, Berin Balachandar Gnana Sekar, Andreas Geiger, Bastian Leibe.
[3] "Video Instance Segmentation" Linjie Yang and Yuchen Fan.
[4] "MaskRNN: Instance Level Video Object Segmentation" Yuan-Ting Hu, Jia-Bin Huang, and Alexander G. Schwing.

---

### Decision · Program_Chairs · 2019-12-19

**Decision:**

Reject

**Comment:**

This paper proposes a spatio-temporal embedding loss for video instance segmentation. The proposed model (1) learns a per-pixel embedding such that the embeddings of pixels from the same instance are closer than embeddings of pixels from other instances, and (2) learns depth in a self-supervised way using a photometric reconstruction loss which operates under the assumption of a moving camera and a static scene. The resulting loss is a weighted sum of these attraction, repulsion, regularisation and geometric view synthesis losses.
The reviewers agree that the paper is well written and that the problem is well motivated. In particular, there is consensus that the 3D geometry and 2D instance representation should be considered jointly. However, due to the lack of technical novelty, the complexity of the final model, and the issues with the empirical validation of the proposed approach, we feel that the work is slightly below the acceptance bar.